# TSMGen: Target-Specific Molecule Generation via Higher-Order Structural Dependencies and Context-Aware Bidirectional Fusion

**Yaoyu Chen** [1]  **Xiaoli Lin** [1 2]  **Ziyi Gong** [1]  **Jun Pang** [1 2]

## Abstract

Efficiently designing high-quality molecules targeting disease-relevant targets is a critical challenge. Most existing methods can capture pairwise amino acid relations, neglecting the higher-order relations among multiple amino acids. This paper proposes a target-specific molecule generation framework, namely TSMGen, to comprehensively capture the local and global structural information of the protein pocket by modeling higher-order spatial dependencies both at the atomic and the amino acid levels. Furthermore, we design a context-aware bidirectional fusion module to learn the more detailed structural information about the protein pocket. This module simultaneously attends to features from both the protein pocket and the molecule, fully leveraging the structural information from both to optimize the generation process of targeted molecules, thereby enhancing the quality of generated molecules. Experiments show that TSMGen outperforms state-of-the-art methods in terms of Vina Score, High Affinity, QED, SA and Diversity, and a case study on $\beta$-secretase enzyme further confirms its ability to generate molecules with stronger binding affinity.

## 1. Introduction

Artificial intelligence-assisted drug design has become a hot topic in drug development research, especially the use of generative deep learning models for novel drug development (Tang et al., 2024). Deep learning models leverage their powerful data processing capabilities and automatic feature extraction capabilities to capture and learn the structural features of known molecules (Chen et al., 2024b; Lin et al., 2024; Simon & Zou, 2025), generating new molecular structures with potential drug activity (Catacutan et al., 2024; Gangwal et al., 2024; Shi et al., 2025). Currently, there are a large number of deep learning-based molecular generation models (Wu et al., 2026; Wang et al., 2025b; Lin et al., 2025). Based on whether these models utilize target protein information, they can be divided into two categories: (1) ligand-based molecular generation models. (2) receptor-based molecular generation models.

Ligand-based molecular generation models utilize the structural or sequence information of known ligand molecules as input, without relying on target protein information, to design molecules (Zhang et al., 2024; Li et al., 2023). It is based on the principle that structurally similar molecules have similar properties (Jiang et al., 2024; Krishnan et al., 2021b; Liu et al., 2025). These models convert molecules into mathematical representations (Bonidia et al., 2022; Merkys et al., 2023), learn structural distributions, and encode them into continuous latent spaces (Lavecchia, 2024; Li et al., 2024; Zhang et al., 2025). However, they inherently lack explicit target protein information. When designing target-specific drugs, such models require large ligand datasets for fine-tuning, which is challenging for rare or emerging diseases where ligand data are scarce (Krishnan et al., 2021a).

Receptor-based molecular generation incorporates protein structural information to constrain the design process (Wang et al., 2024). Protein pocket features guide the generation of molecules structurally complementary to the binding site (Zhao & Bourne, 2022). Since proteins are complex macromolecules (Ni et al., 2024; Chen et al., 2024a), accurately modeling pocket structure is critical. Despite recent advances, challenges remain in two areas: (1) adequately describing higher-order structural dependencies within protein pockets. (2) effectively integrating pocket and molecular information during the generative process.

To address these challenges, this paper proposes a targeted molecular generation method based on higher-order structural dependencies and context-aware bidirectional fusion. Protein pockets are represented as hypergraphs to

[1]School of Computer Science and Technology, Wuhan University of Science and Technology, Wuhan, Hubei, China [2]Hubei Province Key Laboratory of Intelligent Information Processing and Real-Time Industrial System, Wuhan University of Science and Technology, Wuhan, Hubei, China. Correspondence to: Xiaoli Lin <linxiaoli@wust.edu.cn>.

*Proceedings of the 43rd International Conference on Machine Learning*, Seoul, South Korea. PMLR 306, 2026. Copyright 2026 by the author(s).

capture higher-order residue relationships effectively modeling the global topological context and spatial organization of residues. Combined with atomic-level graphs, we extract fine-grained pocket structure to incorporate precise local geometric details and physicochemical properties. Furthermore, a bidirectional gated cross-attention module integrates pocket-molecule features to supplement interaction-aware semantic information, ensuring the generated molecules are dynamically compatible with the pocket environment, which significantly enhances target specificity.

The major contributions of this work are summarized as follows:

- We proposed a target-specific molecule generation framework with context-aware bidirectional fusion. It can capture complex higher-order structural dependencies within protein pockets that helps optimize the performance of targeted molecule generation.

- We fused residue-level and atomic-level structure features from protein pockets to capture the local and global structural information. We also developed a context-aware bidirectional fusion module that incorporates gated cross-attention. It fuses two modal embeddings from the protein pocket and the molecule. This fused feature serves as a condition for the generation targeted molecules.

- Experiments conducted on public datasets, demonstrate stronger performance across multiple drug-likeness metrics. The results indicate that TSMGen is capable of generating molecules with favorable affinity and target specificity.

## 2. Related Work

### 2.1. Ligand-based Molecular Generation

Ligand-based molecular generation models design molecules without using target protein information. They rely on structural or sequence data from existing ligands, learn molecular distributions, extract representative features, and encode molecules into low-dimensional continuous spaces.

**VAE-based Methods.** Gómez-Bombarelli et al. (2018) pioneered the application of VAEs to molecular generation with the ChemVAE model. Later, several improved graph-based VAE variants were developed, including ConfVAE (Xu et al., 2021b), PS-VAE (Kong et al., 2022), and MoVAE (Lin et al., 2023).

**RNN-based Methods.** Li et al. (2020); Grisoni et al. (2020) modeled molecular sequences to learn chemical grammar and generate new molecules. Furthermore, Moret et al.

(2023) fine-tuned pretrained RNN generators for target-specific data, while Goel et al. (2021); Hu et al. (2023) employed reinforcement learning to optimize physicochemical properties.

**Transformer-based Methods.** Transformers have demonstrated strong capabilities in molecular sequence modeling. Monteiro et al. (2023) designed biologically active AA2AR-targeting molecules, while Mao et al. (2023) generated antiviral molecules with favorable physicochemical properties. Large-scale language model variants have also been applied to molecular generation Haroon et al. (2023); Ye (2024) trained GPT-like models capable of producing drug-like molecules with enhanced lipophilicity.

Although the above ligand-based methods have made some progress in the field of molecular generation, such methods can only generate new molecular structures that comply with chemical rules, and lack specific information about disease targets.

### 2.2. Receptor-based Molecular Generation

Receptor-based approaches incorporate structural information from target proteins to guide molecule design. Protein pocket modeling enables generating molecules complementary to the binding site. Accurate structural representation is vital because proteins exhibit complex 3D conformations and nonlinear residue interactions.

**Sequence-based Molecular Generation Methods.** Grechishnikova (2021); Qian et al. (2022) represent target proteins as amino acid sequences and convert the molecular generation task into a translation task from protein amino acid sequences to molecular sequences using the Transformer model, directly generating target-specific molecules for the disease based on the amino acid sequences of the disease targets. Chen et al. (2023) constructed embedded representations of target amino acid sequences using pre-trained protein language models and combined them with GAN networks to generate molecules with high affinity for binding to target proteins. However, the above method of representing proteins as amino acid sequences does not fully take into account the spatial structural information of proteins and the higher-order relationships between multiple amino acids.

**3D Pocket-based Autoregressive Methods.** Xu et al. (2021a) took into account the structural information of the target protein and proposed representing the three-dimensional information of the protein pocket as a pocket descriptor based on the Coulomb matrix, which was then used as a condition to constrain the molecular generation process. Ragoza et al. (2022); Weller & Rohs (2024) represent protein pockets as voxelized atomic density grids, predict the probability density of ligand molecules appearing

in protein pockets, and then use this to gradually generate atoms and assemble them into molecules. Peng et al. (2022) modeled the positional relationships between atoms and bonds in protein pocket spaces based on isometric graph neural networks, and then gradually generated molecules. Peng et al. (2022) modeled the positional relationships between atoms and bonds in protein pocket spaces based on isometric graph neural networks, and then gradually generated molecules. Subsequent work, such as that by Zhang & Liu (2023); Zhang et al. (2023), introduced fragment-level autoregressive construction. However, the above autoregressive method based on protein pocket structure often produces chemically ineffective intermediates or unreasonable molecular fragments during the gradual generation of molecules.

**Diffusion-based Models.** To address these issues, Ghorbani et al.; Huang et al. (2024) employed diffusion models that generate complete molecules during denoising, thereby avoiding invalid intermediate states. However, since the diffusion model is based on noise diffusion and reconstruction, it ignores certain physical rules, resulting in certain limitations when guiding molecular generation.

## 3. Methods

We propose a molecular generation framework, TSMGen, focused on generating candidate molecules that can efficiently bind to protein pockets. This framework consists of five components. First, it uses residue-level hypergraphs to capture residue-level many-body interactions and performs fine-grained processing of pocket binding environments at the atomic-level. Next, it encodes the molecules. Specifically, the three-dimensional structures of candidate molecules are converted into standardized SMILES strings to retain core topological connectivity, which are then mapped into a continuous vector space via a sequence encoder to generate dense molecular feature representations. Then, it uses context-aware bidirectional fusion technology to achieve bidirectional fusion between pocket and molecular features. Finally, it generates molecules targeted at protein pockets. The overall framework is illustrated in Figure 1.

### 3.1. Residue-Level Hypergraph Construction for Protein Pockets

The structural and biological functions of protein pockets are intrinsically linked to their amino acid composition. To capture this complexity, we define each protein pocket as a hypergraph $G = (V, E, X)$. Conceptually, as illustrated in Figure 2, the physical 3D structure of the pocket is visually mapped into a topological representation, where nodes represent individual amino acids and hyperedges capture local spatial clusters of these residues. Computationally, as

---

**Algorithm 1** Protein Pocket Hypergraph Construction

**Input:** Protein pocket 3D structure object $P$
**Output:** Hypergraph $G = (V, E, X)$ where:
    $V$: Amino acid node set
    $E$: Hyperedge set
    $X$: Node feature set
**Function** BuildHypergraph($P$)
Obtain amino acid set $V_{\text{set}} = \{v_1, v_2, \ldots, v_n\}$ from $P$
Initialize $V \leftarrow V_{\text{set}}, E \leftarrow \emptyset, X \leftarrow \emptyset$
**for** $i = 1$ **to** $n$ **do**
    Let $v_i$ be the $i$-th amino acid in $V_{\text{set}}$
    Get $C_\alpha$ atom coordinate $v_{\text{xyz},i} \leftarrow \text{coordinates}(v_i)$
    Initialize hyperedge $e_i \leftarrow \{v_i\}$
    Extract node features $x_i \leftarrow \phi(v_i)\; X \leftarrow X \cup \{x_i\}$
    **for** $j = 1$ **to** $n$ **do**
        **if** $i \neq j$ **then**
            Let $v_j$ be the $j$-th amino acid in $V_{\text{set}}$
            Get $C_\alpha$ atom coordinate:
                $v_{\text{xyz},j} \leftarrow \text{coordinates}(v_j)$
            Compute Euclidean distance:
                $distance_{ij} \leftarrow \|v_{\text{xyz},i} - v_{\text{xyz},j}\|_2$
            **if** $distance_{ij} \leq 5$ **then**
                $e_i \leftarrow e_i \cup \{v_j\}$
            **end if**
        **end if**
    **end for**
    $E \leftarrow E \cup \{e_i\}$
**end for**
**Return** $G \leftarrow (V, E, X)$
**End Function**

---

detailed in Algorithm 1, this construction relies on specific geometric rules: the algorithm defines the hyperedge set $E = \{e_1, e_2, \ldots, e_n\}$ by calculating the Euclidean distance between the $C_\alpha$ atoms of the residues, grouping a central node and all its neighboring nodes within a 5 Å distance threshold into a single hyperedge.

Within this hypergraph, $V = \{v_1, v_2, \ldots, v_n\}$ is the set of amino acid nodes, where $n$ is the number of residues. The node feature matrix is defined as $X \in \mathbb{R}^{n \times d_p}$, where $d_p$ is the feature dimension. Each amino acid node $v_i$ incorporates biochemical and structural features represented as $\mathbf{x}_i = [\mathbf{x}_i^{\text{type}}; \mathbf{x}_i^{\text{ss}}; \mathbf{x}_i^{\text{spatial}}] \in \mathbb{R}^{d_p}$, where $d_p = 32$.

### 3.2. Residue-Level Structure Representation of Protein Pockets

After constructing the protein pocket hypergraph $G$, we process it using a Hypergraph Neural Network (HGNN) to iteratively aggregate and update node features, followed by a pooling mechanism to obtain the global representation $G_b$. Unlike traditional graph neural networks that propagate information through pairwise connections, HGNN performs

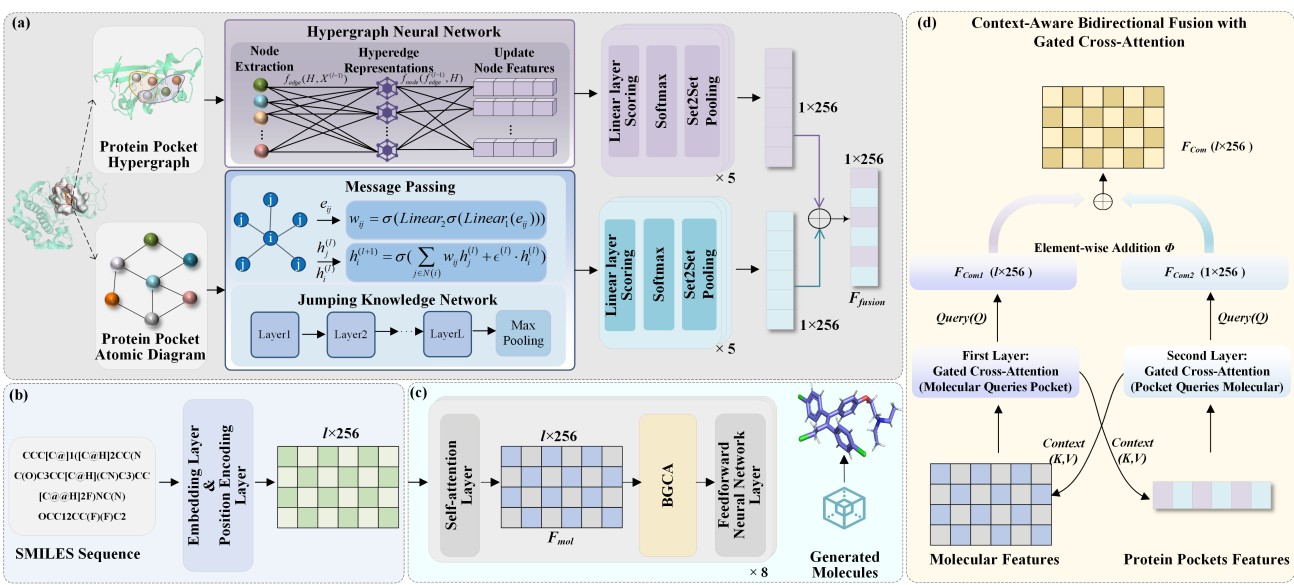

*Figure 1.* Framework of the proposed TSMGen. (a) Residue-level hypergraph and Atomic-level structure representation of protein pockets. (b) Feature representation of molecules. (c) Molecule generation module. (d) Context-aware bidirectional fusion.

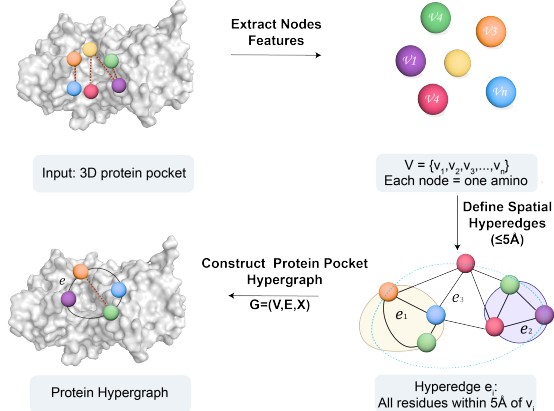

*Figure 2.* Protein hypergraph construction process.

message passing in a node-hyperedge-node manner. Each propagation layer consists of two sequential operations:

**Node-to-Hyperedge Aggregation:** Node features connected to each hyperedge are aggregated to form hyperedge representations:

$$F_{\text{edge}}^{(l-1)} = f_{\text{edge}}\left(H, X^{(l-1)}\right) \tag{1}$$

where $H \in \{0,1\}^{n \times |E|}$ is the hypergraph incidence matrix, $X^{(l-1)} \in \mathbb{R}^{n \times d_p}$ denotes the node feature matrix at layer $l-1$, and the function $f_{\text{edge}}$ represents the hyperedge aggregation operation.

**Hyperedge-to-Node Update:** Hyperedge features are then

propagated back to update node embeddings:

$$X^{(l)} = f_{\text{node}}\left(F_{\text{edge}}^{(l-1)}, H\right) \tag{2}$$

where $F_{\text{edge}}^{(l-1)} \in \mathbb{R}^{|E| \times d_p}$ denotes the intermediate hyperedge features with dimensionality $d_p = 256$, and the function $f_{\text{node}}$ represents the node feature update operation.

This dual-phase propagation captures higher-order, multi-residue interactions within the pocket, enabling the model to learn structural patterns that cannot be expressed through simple pairwise graphs.

To obtain the global pocket representation, we employ Set2Set pooling, which uses an attention-driven aggregation scheme to iteratively refine the global descriptor. At each iteration $t$, node features are aggregated as:

$$r_t = \sum_{i=1}^{N} \alpha_{i,t} \mathbf{x}_i \tag{3}$$

$$\alpha_{i,t} = \text{softmax}\left(x_i^\top W h_t\right)$$

where $r_t \in \mathbb{R}^{d_p}$ denotes the attention-weighted node aggregation, and $\alpha_{i,t}$ represents the attention coefficient for node $i$. The vector $h_t \in \mathbb{R}^{d_p}$ is an internal query state that guides attention at iteration $t$.

The final global representation is obtained by concatenating the last-iteration query state and aggregated features:

$$G_b = h_T \parallel r_T \tag{4}$$

where $T$ denotes the number of iterations and $\parallel$ denotes feature concatenation.

Through iterative refinement and attention-based aggregation, the Set2Set pooling captures long-range dependencies among residues and integrates both contextual information (contained in $h_T$) and structural cues (captured by $r_T$). The resulting representation $G_b$ provides a comprehensive residue-level characterization of the protein pocket.

### 3.3. Atomic-Level Structure Representation of Protein Pockets

The structure of protein pockets consists of atoms and chemical bonds, which can be represented as a graph with atoms as nodes and chemical bonds as edges. The node features of the protein pocket atomic graph include three-dimensional atomic coordinates (absolute coordinates, relative coordinates in Cartesian coordinate system) and some physicochemical properties. The edge features include types of chemical bonds, which describe the relationship between each atom and its neighboring atoms.

Multi-layer message passing with jumping knowledge network is used to aggregate and update node features. The message passing block aggregates neighbor node features and propagates them as messages between nodes to achieve node feature aggregation and updating. The jumping knowledge network integrates node features from multiple message passing layers, effectively preserving important node features in the graph.

In each layer of message passing, the update of node features depends on the features of their neighbor nodes and edge features. During the feature update of a single node, the weight $w_{ij}$ between node $i$ and neighbor node $j$ is first calculated based on edge features, and the neighbor node features are weighted according to this weight to obtain messages. The weight $w_{ij}$ is defined as:

$$w_{ij} = \sigma \left( \text{Linear}_2 \left( \sigma \left( \text{Linear}_1 \left( e_{ij} \right) \right) \right) \right) \quad (5)$$

where $w_{ij}$ is the edge weight between node $i$ and node $j$, $e_{ij}$ is the edge feature between node $i$ and node $j$, and $\text{Linear}_1$ and $\text{Linear}_2$ represent linear layers.

Subsequently, node $i$ aggregates the information from all its neighbor nodes and updates its features accordingly. The update process is defined as :

$$h_i^{(l+1)} = \sigma \left( \sum_{j \in \mathcal{N}(i)} w_{ij} h_j^{(l)} + \epsilon^{(l)} \cdot h_i^{(l)} \right) \quad (6)$$

where $h_i^{(l)}$ and $h_i^{(l+1)}$ represent the features of node $i$ at $l$-th layer and $(l+1)$th layer respectively, $\mathcal{N}(i)$ represents the set of neighbor nodes of node $i$, $\epsilon^{(l)}$ represents the learnable parameter at $t$-th layer, and $\sigma$ represents the activation function.

After completing multi-layer message passing, to effectively integrate node features from different layers, a jumping knowledge network is used to fuse node features from each layer. The jumping knowledge network allows selecting the most appropriate network layer for individual nodes, and aggregates node features from all layers through max pooling. It effectively preserves node information from different layers and reduces information loss. The max pooling process can be expressed as:

$$h_i^{\text{JK}} = \text{max\_pooling} \left( h_i^{(1)}, h_i^{(2)}, \ldots, h_i^{(L)} \right) \quad (7)$$

where $h_i^{\text{JK}}$ represents the feature of node $i$ after the jumping knowledge network, $L$ represents the number of layers in the jumping knowledge network, and $h_i^{(L)}$ represents the feature of node $i$ at layer $L$.

Finally, the Set2Set pooling is used to read out the complete graph structure feature representation from node features, obtaining an atomic graph feature vector with feature dimension of $1 \times 256$.

### 3.4. Feature Representation of Molecules

After preprocessing the dataset to remove complex molecular information (including salts and stereochemistry), we convert the 3D molecular structures into SMILES strings. The notation system prioritizes atomic connectivity over spatial configuration, as formalized by the SMILES specification. The adoption of SMILES sequences for representing drug molecules offers significant advantages in feature fusion with protein pocket representations. SMILES provides a standardized linear notation that preserves essential molecular graph topology while eliminating structural complexities like stereochemistry and salts through preprocessing. This creates a consistent, simplified representation compatible with sequence-based deep learning architectures.

### 3.5. Context-Aware Bidirectional Fusion

We propose a **Context-Aware Bidirectional Fusion with Gated Cross-Attention (CABF)** module to enhance mutual feature selection and deep fusion between molecular and protein pocket representations. CABF employs a dual-layer gated cross-attention mechanism, as illustrated in Figure 1.

**First Layer:** Molecular features $F_M \in \mathbb{R}^{l \times 256}$, where $l$ is molecular sequence length. These features generate query vectors to extract protein pocket information relevant to molecular characteristics. A gating mechanism yields the fused representation:

$$F_{com_1} \in \mathbb{R}^{l \times 256} \quad (8)$$

**Second Layer:** Protein pocket features $F_P \in \mathbb{R}^{1 \times 256}$ generate query vectors to extract molecular features relevant to

pocket characteristics. The feature representations obtained through the gate mechanism are as follows:

$$F_{com_2} \in \mathbb{R}^{1 \times 256} \tag{9}$$

The final fused feature $F_{com} \in R^{l \times 256}$ is obtained by repeating $F_{com_2}$ along the sequence dimension and element-wise addition:

$$F_{\text{com}} = \phi\Big(F_{\text{com}_1}, R(F_{\text{com}_2}, l)\Big) \tag{10}$$

where $\phi$ denotes the addition operator and $\mathcal{R}$ the sequence-length replication operation.

### 3.6. Molecule Generation Module

Our framework achieves de novo pocket conditioning through a clearly defined input-output interface. Specifically, during the training phase, the conditional molecular generator receives both molecular features and structural protein pocket features as inputs. By employing a gated cross-attention mechanism, it fuses the molecular data with the higher-order spatial information processed by the pocket representation module. This tightly integrated condition effectively steers the SMILES-based autoregressive process, thereby enabling the generation of novel molecules tailored to specific protein pockets.

During the model training phase, the cross-entropy loss function is used to measure the difference between the model's predicted distribution and the true distribution. The model parameters are updated through backpropagation to minimize the difference between them. The loss function is defined as:

$$\text{Loss} = -\frac{1}{N} \sum_{i=1}^{N} \sum_{c=1}^{M} y_{ic} \log(p_{ic}) \tag{11}$$

In this context, $N$ denotes the batch size, and $M$ represents the total number of categories, that is, the number of all different types of features that may appear in the molecular sequence. $p_{ic}$ denotes the probability that the model predicts the $i$-th sample belongs to category $c$. $y_{ic}$ is used to reflect whether the true category of the $i$-th sample belongs to $c$. If the $i$-th sample belongs to category $c$, the value of $y_{ic}$ is 1, otherwise it is 0.

## 4. Experiment

### 4.1. Preprocessing of Dataset

The CrossDocked dataset provides protein-ligand pairs with separated protein pocket structures (PDB format) and docked ligand structures (SDF format). The initial dataset contained 22.5 million docked poses. Luo et al. (2021) filtered this set by removing poses with ligand RMSD > 1 Å and molecules failing RDKit standardization, resulting in

approximately 160,000 refined pairs. Proteins were then clustered at 30% sequence identity using MMseqs2. For typical usage, 100,000 pairs were randomly sampled for training, while 100 pairs from held-out clusters formed the test set. For our specific requirements of representing protein pockets as hypergraphs and ligands as SMILES strings, we performed additional preprocessing: (1) Ligand Filtering: Ligand SDF files were converted to canonical SMILES strings using OpenBabel. Pairs where this conversion failed were discarded. (2) Protein Pocket Filtering: Protein pockets were processed for hypergraph construction. Pockets that failed construction (e.g., due to amino acid mutations or hetero domains) were removed, along with their corresponding ligands. This preprocessing resulted in a final dataset of 14,822 protein-ligand pairs (14,822 protein pockets and 14,822 ligand molecules).

### 4.2. Implementation Details

The model was implemented in Python 3.8 using PyTorch 1.11 and PyG (PyTorch Geometric) 2.4. Experiments were conducted on an Ubuntu 20.04 system with an NVIDIA GeForce RTX 3090 GPU (24GB VRAM), 64GB CPU RAM, and CUDA 11.3 with cuDNN 8.4 acceleration. Training utilized the Adam optimizer with an initial learning rate of 0.0003 and weight decay (L2 regularization) of 0.0005. A dynamic learning rate scheduler reduced the rate by a factor of 0.5 upon validation loss plateau, continuing until reaching the minimum learning rate or training completion. The model was trained for 30 epochs with a batch size of 32. The hypergraph neural network architecture consisted of 2 layers. The jumping knowledge network consisted of 5 layers. Hyperparameter sensitivity analysis details are in Appendix A.

### 4.3. Metrics

To comprehensively evaluate the quality and viability of the generated compounds, a systematic assessment framework is required. This paper uses several common evaluation metrics in targeted molecular generation to assess model performance, including Vina Score, High Affinity, SA Score, QED, LogP, Lipinski, and Diversity. Among these, the Vina Score assesses the binding stability and strength between the generated molecules and the protein pocket. A lower value indicates greater stability and strength. High Affinity indicates the proportion of generated molecules exhibiting stronger binding affinity than the reference ligand in the test set. QED measures the likelihood of a molecule becoming a drug by calculating its similarity to known drugs. SA assesses the ease of synthesis during drug development. LogP measures the lipophilicity or hydrophilicity of a substance. Lipinski evaluates whether a molecule complies with the basic rules for oral drugs, while Diversity measures the structural variety among the generated molecules.

*Table 1.* Ablation study results of different features. (**Best**, Second Best)

| Model | Vina Score (↓) | High Affinity (↑) | QED (↑) | SA (↑) | LogP | Lipinski (↑) | Diversity (↑) |
|---|---|---|---|---|---|---|---|
| w/o ALF | -7.509 | 0.635 | 0.624 | 0.755 | 3.030 | 4.913 | 0.816 |
| w/o RLF | -7.435 | 0.621 | 0.578 | 0.745 | 2.851 | 4.824 | **0.822** |
| w/o MolF | -7.510 | **0.645** | 0.647 | 0.754 | 2.958 | 4.942 | 0.819 |
| w/o CABF | -7.501 | 0.635 | 0.639 | 0.754 | 2.887 | 4.938 | 0.819 |
| TSMGen | **-7.642** | 0.631 | **0.659** | **0.762** | 2.883 | **4.966** | 0.753 |

*Table 2.* Performance comparison of TSMGen and other different methods. (**Best**, Second Best)

| Model | Vina Score (↓) | High Affinity (↑) | QED (↑) | SA (↑) | LogP | Lipinski (↑) | Diversity (↑) |
|---|---|---|---|---|---|---|---|
| LiGAN (Ragoza et al., 2022) | -6.032 | 0.194 | 0.365 | 0.615 | -0.140 | 4.002 | 0.667 |
| DrugGPS (Zhang & Liu, 2023) | -7.276 | 0.565 | 0.613 | 0.743 | 1.134 | 4.917 | 0.681 |
| FLAG (Zhang et al., 2023) | -7.247 | 0.580 | 0.495 | 0.745 | 0.630 | 4.943 | 0.704 |
| AutoFragDiff (Ghorbani et al.) | -7.450 | - | 0.450 | 0.620 | - | - | 0.690 |
| PMDM (Huang et al., 2024) | -7.572 | 0.628 | 0.594 | 0.611 | 0.301 | **4.975** | 0.709 |
| DTF-diffusion (Wang et al., 2025a) | -6.820 | - | 0.446 | 0.725 | - | 4.403 | 0.740 |
| AMDIff (Li et al., 2025) | -7.466 | - | 0.471 | 0.681 | - | - | 0.672 |
| TSMGen | **-7.642** | **0.631** | **0.659** | **0.762** | 2.883 | 4.966 | **0.753** |

## 4.4. Ablation Study

We carried out a comprehensive ablation study to quantify the effectiveness of different modules, including the atomic-level feature (ALF), residue-level feature (RLF), molecule feature (MolF), and cross-attention binding feature (CABF). As detailed in Table 1, the removal of any individual component leads to a noticeable degradation in model capability. In particular, the variant without residue-level features (w/o RLF) exhibits the poorest performance in both binding stability and drug-likeness, underscoring the critical role of residue information in capturing pocket geometry.

By effectively integrating all modules, TSMGen demonstrates superior performance in key metrics for drug molecule generation. Its predicted molecular binding energy (Vina Score: -7.642) outperforms those of all ablated models, with binding strength enhanced by 1.8% to 2.8% compared to the ablated models. Furthermore, both the probability of drug-likeness (QED: 0.659) and synthetic feasibility (SA: 0.762) rank highest, confirming the model's ability to generate practical pharmaceutical candidates. Lipinski's rule compliance reaches 4.966 (approaching the ideal score of 5), indicating optimal oral absorption potential. While some ablated models exhibit slightly higher randomness, TSMGen (Diversity: 0.753) achieves a favorable balance, maintaining high structural novelty without compromising chemical validity. TSMGen leads in comprehensive drug-likeness indicators.

## 4.5. Comparison

Table 2 presents a comprehensive quantitative evaluation of TSMGen against a broad range of state-of-the-art baselines.

As observed, TSMGen establishes a new state-of-the-art across the core metrics of drug molecule generation. In terms of binding capabilities, it achieves an optimal Vina Score of -7.642, surpassing the second-best method, PMDM (-7.572), by a clear margin. Notably, TSMGen also secures the highest High Affinity score (0.631), empirically validating its superior ability to generate molecules with binding potentials exceeding those of reference ligands.

Additionally, the model demonstrates exceptional performance in physicochemical properties: its QED value reaches 0.659, significantly outperforming the runner-up DrugGPS (0.613). Moreover, the SA score (0.762) exceeds the second best (FLAG, 0.745), indicating that the generated candidates possess enhanced synthetic feasibility for practical wet-lab development. Regarding the Lipinski score, TSMGen maintains a high value of 4.966, remaining statistically comparable to the top-performing baseline PMDM (4.975). Furthermore, TSMGen achieves a Diversity score of 0.753, surpassing both DTF-diffusion (0.740) and PMDM (0.709). These experimental results demonstrate the superior performance of TSMGen, indicating its strong competitiveness and promising potential for real-world applications in structure-based drug design.

## 4.6. Case Study

Figure 3 visualizes the top four generated candidates for $\beta$-secretase enzyme (PDB ID: 7D5U) through their 2D molecular graphs, SMILES sequences, and 3D docking poses. This multi-view visualization highlights the precise geometric complementarity of the ligands within the protein pocket. Notably, the optimal candidates achieve a binding energy of

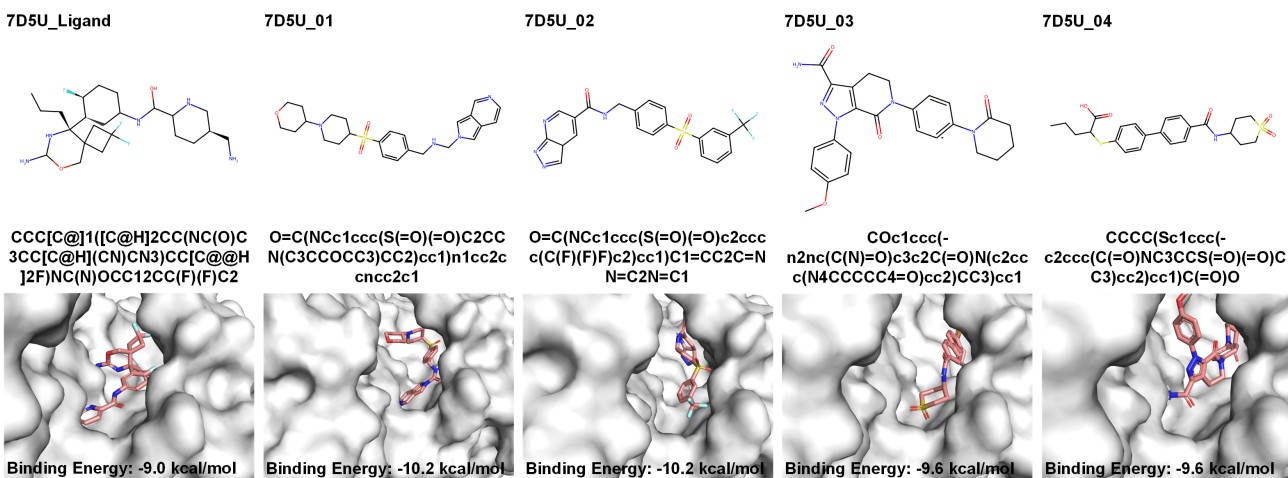

7D5U_Ligand     7D5U_01     7D5U_02     7D5U_03     7D5U_04

CCC[C@]1([C@H]2CC(NC(O)C3CC[C@H](CN)CN3)CC[C@@H]2F)NC(N)OCC12CC(F)(F)C2

O=C(NCc1ccc(S(=O)(=O)C2CCN(C3CCOCC3)CC2)cc1)n1cc2cncc2c1

O=C(NCc1ccc(S(=O)(=O)c2cccc(C(F)(F)F)c2)cc1)C1=CC2C=NN=C2N=C1

COc1ccc(-n2nc(C(N)=O)c3c2C(=O)N(c2ccc(N4CCCCC4=O)cc2)CC3)cc1

CCCC(Sc1ccc(-c2ccc(C(=O)NC3CCS(=O)(=O)C3)cc2)cc1)C(=O)O

Binding Energy: -9.0 kcal/mol   Binding Energy: -10.2 kcal/mol   Binding Energy: -10.2 kcal/mol   Binding Energy: -9.6 kcal/mol   Binding Energy: -9.6 kcal/mol

*Figure 3.* Visualization of four ligand molecules binding to the 7D5U protein target.

-10.2 kcal/mol, significantly surpassing the reference ligand (-9.0 kcal/mol).Detailed structural and stability analyses of the top ten candidates are provided in Appendix B.

To evaluate the overall quality of the generated library, we analyzed both the binding energy distribution and conformational stability, as visualized in Figure 4(a). demonstrates that the TSMGen model effectively shifts the distribution towards high-affinity interactions, with a substantial number of candidates surpassing the native ligand's threshold (indicated by the red dashed line) and the distribution tail extending significantly into a more favorable energy range. Furthermore, Figure 4(b) illustrates the RMSD distribution relative to the best-scored pose, where the primary cluster of poses falls within a low RMSD range. This indicates that the generated molecules tend to form defined and stable binding modes rather than adopting random, unstable orientations.

*Table 3.* Docking binding site information for the four molecules with the best docking binding energy for 7D5U.

| No. | Binding Site Residue | Distance ($\mathring{A}$) |
|---|---|---|
| 7D5U_01 | PHE-124 | 2.1 |
| 7D5U_01 | TRP-92 | 2.8 |
| 7D5U_01 | SER-51 | 2.5 |
| 7D5U_02 | ASN-53 | 3.4 |
| 7D5U_02 | ASN-53 | 3.1 |
| 7D5U_02 | LEU-142 | 2.9 |
| 7D5U_02 | LEU-142 | 3.2 |
| 7D5U_03 | ASN-53 | 2.4 |
| 7D5U_03 | SER-52 | 1.9 |
| 7D5U_03 | TRP-131 | 2.1 |
| 7D5U_04 | GLY-243 | 2.4 |

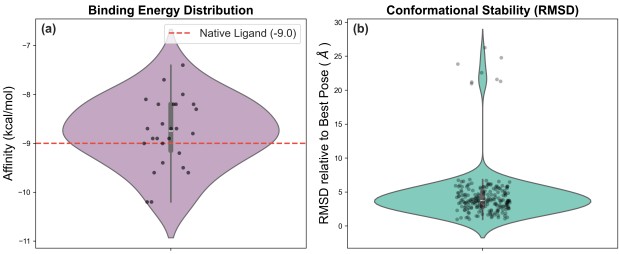

*Figure 4.* Statistical evaluation of the docking simulations. (a) Binding energy distribution of the generated molecules. (b) RMSD distribution of alternative docking poses.

To quantify the specific interactions contributing to this stability, Table 3 details the hydrogen bond information for the top candidates, including the interacting amino acid residues and bond distances. Complementing this data, Figure 5 illustrates the specific docking state of the generated molecule with the target protein. The purple sticks indicate the key amino acids, the light pink sticks indicate the generated

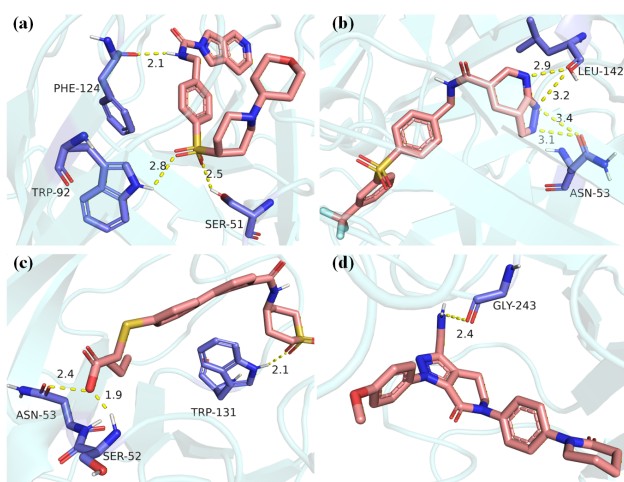

*Figure 5.* Docking diagram of the four molecules with the best docking binding energy for 7D5U.

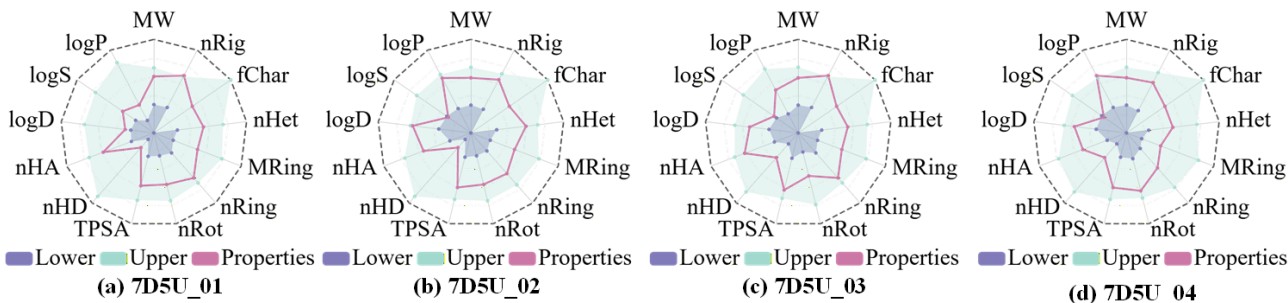

*Figure 6.* Property diagrams of the four molecules with the best docking binding energy for 7D5U.

molecules, and the yellow dashed lines represent hydrogen bond interactions between the proteins and the molecules.

The observed distances (1.9Å to 3.4Å) indicate strong interactions and stable binding poses. These combined results confirm that the generated molecules possess both the high binding affinity and structural stability required for potential inhibitors.

Furthermore, we evaluated the drug-likeness of these top candidates. In Figure 6, the dark purple background indicates the lower limit for each property, the light green background indicates the upper limit for each attribute, and the light purple outline indicates the actual value of each property of the molecule. All properties of the four molecules fall within the expected range, indicating that the generated molecules possess desirable physicochemical properties and structural characteristics.

In addition to structural, energetic, and physicochemical properties, the practical synthesizability of compounds designed by deep generative models remains a key challenge in AI-aided drug discovery. To further assess the practical synthesizability of the molecules generated by our proposed TSMGen framework, we conducted a retrosynthetic analysis. Specifically, we utilized the inverse synthesis planning tool, AiZynthFinder, to evaluate the 50 generated molecules for the 7D5U target presented in our case study.

The evaluation results demonstrate that 30% of the generated molecules are predicted to be synthesizable in 3 to 4 steps, 60% are predicted to be synthesizable in 5 to 6 steps, and only 10% require 7 or more steps. These findings indicate that the majority of the molecules generated by TSMGen maintain a high degree of synthetic accessibility and require a reasonable number of reaction steps for practical execution.

## 5. Conclusion

We propose a target-specific molecular generation model, namely TSMGen, that integrates higher-order structural dependencies to optimize the quality of molecules. TSMGen combines the residue-level representation and atomic-level representation to capture local and global structure information of protein pockets. In addition, we designed a context-aware bidirectional fusion module with bidirectional gated cross-attention, which enables deep mutual integration of protein pocket features and molecular features to enhance the targeting of generated molecules. Experiments demonstrate that TSMGen generates molecules with higher binding affinity and superior chemical quality, showing that it holds potential for the drug design task. In future work, we will integrate dynamic pocket flexibility modeling and mechanical descriptors to enhance generative accuracy for multi-targets.

## Acknowledgments

The authors thank the members of the Machine Learning and Artificial Intelligence Laboratory, School of Computer Science and Technology, Wuhan University of Science and Technology, for helpful discussions during laboratory seminars. This work was supported in part by the Hubei Province Natural Science Foundation of China (No.2024AFB865) and the National Natural Science Foundation of China (No.62372342, 61972299).

## Code and Data Availability

The source code and datasets are available at `https://github.com/Chenyy555/TSMGen`.

## Impact Statement

This paper presents work whose goal is to advance the field of Machine Learning, specifically by enhancing AI-driven drug discovery. Our approach has the potential to significantly accelerate the identification of viable therapeutic candidates, thereby reducing the time and cost of pharmaceutical development. From a broader perspective, this contributes to the broader goal of leveraging AI to address critical healthcare challenges and improve public health.

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

# A. Hyperparameter Sensitivity Analysis

To evaluate the stability and robustness of the proposed model, we conducted a sensitivity analysis on key hyperparameters: learning rate, Hypergraph Neural Network layers, and Jumping Knowledge Network layers. The detailed experimental results for all configurations are summarized in Table 4.

## A.1. The Impact of Learning Rate

The learning rate is a central factor controlling the magnitude of parameter updates during model training. Its proper selection directly affects the convergence speed, stability, and final performance of the optimization process. A learning rate that is too small slows convergence and significantly increases the number of iterations required. In contrast, an excessively large learning rate results in overly aggressive updates, causing oscillation around the optimal point and preventing stable convergence. We evaluate learning rates of 0.0001, 0.0002, 0.0003, 0.0005, and 0.001. Our experiments indicate that 0.0003 achieves the best overall performance, balancing convergence speed and stability.

## A.2. The Impact of Hypergraph Neural Network Layers

The number of hypergraph neural network layers is a crucial parameter that affects both model performance and the suitability of the architecture for different tasks. Fewer layers reduce computational cost and avoid excessive message propagation, thus mitigating feature oversmoothing, but they may fail to capture global topological relationships and long-range dependencies in the hypergraph. Increasing the number of layers enables the model to aggregate information from multi-hop hyperedges and better capture global structure. However, deeper architectures risk homogenizing node features, reducing discriminability, and increasing computational complexity and overfitting. Therefore, choosing an appropriate depth requires balancing expressive power and model stability. We experiment with 1, 2, 3, 5, and 10 layers. The optimal configuration uses 2 layers.

## A.3. The Impact of Jumping Knowledge Network Layers

The number of layers in the Jumping Knowledge Network is a key factor influencing its capability to extract topological information, the quality of aggregated representations, and training stability. Too few layers limit the model's ability to integrate multi-scale features spanning from local to global structures. Conversely, too many layers may introduce redundancy and noise, leading to unnecessary computational overhead and increased memory consumption. We test Jumping Knowledge Networks with 1, 2, 3, 5, and 10 layers. The configuration with 5 layers achieves the best performance.

*Table 4.* Results of hyperparameter sensitivity analysis. Note that the rows for LR=0.0003, HGNN Layers=2, and JK Layers=5 represent the same optimal model configuration, hence the identical results. (**Best**, Second Best)

| Hyperparameter | Value | Vina (↓) | High Aff (↑) | QED (↑) | SA (↑) | LogP | Lipinski (↑) | Div. (↑) |
|---|---|---|---|---|---|---|---|---|
| | 0.0001 | -6.873 | 0.582 | 0.567 | 0.692 | 3.125 | 4.321 | 0.678 |
| | 0.0002 | -7.215 | 0.601 | 0.645 | 0.728 | 2.957 | 4.883 | 0.712 |
| Learning Rate | 0.0003 | **-7.642** | **0.631** | **0.659** | **0.762** | 2.883 | **4.966** | **0.753** |
| | 0.0005 | -7.489 | 0.620 | 0.641 | 0.745 | 2.912 | 4.657 | 0.736 |
| | 0.001 | -7.181 | 0.605 | 0.591 | 0.705 | 0.962 | 4.517 | 0.691 |
| | 1 | -6.724 | 0.575 | 0.553 | 0.681 | 3.201 | 4.215 | 0.665 |
| Hypergraph Neural | 2 | **-7.642** | **0.631** | **0.659** | **0.762** | 2.883 | **4.966** | **0.753** |
| Network Layers | 3 | -7.358 | 0.612 | 0.628 | 0.733 | 2.984 | 4.762 | 0.723 |
| | 5 | -7.396 | 0.608 | 0.615 | 0.745 | 2.769 | 4.638 | 0.709 |
| | 10 | -6.943 | 0.589 | 0.582 | 0.698 | 2.634 | 4.402 | 0.684 |
| | 1 | -6.689 | 0.571 | 0.548 | 0.677 | 3.245 | 4.183 | 0.659 |
| Jumping Knowledge | 2 | -7.312 | 0.607 | 0.622 | 0.729 | 3.012 | 4.715 | 0.718 |
| Network Layers | 3 | -7.598 | 0.627 | 0.651 | 0.754 | 2.897 | 4.921 | 0.746 |
| | 5 | **-7.642** | **0.631** | **0.659** | **0.762** | 2.883 | **4.966** | **0.753** |
| | 10 | -7.015 | 0.593 | 0.589 | 0.703 | 2.678 | 4.457 | 0.691 |

# B. Detailed Case Study: 7D5U Molecule Generation

Figure 7 displays the top ten molecular structures ranked by docking binding energy. The best-performing molecule achieved a docking binding energy of −10.2 kcal/mol, surpassing the reference ligand molecule's value of −9.0 kcal/mol. The main text has provided detailed discussions on the top four candidate molecules, while this appendix offers a comprehensive overview of all ten top-ranked molecular structures for supplementary reference.

To further assess the reliability of these top candidates, Figure 8 illustrates the docking energy landscape and conformational stability. The box plots depict the energy distribution of all generated docking poses for each molecule. A compact distribution (short box height) at low energy levels indicates that the molecule consistently converges to a stable high-affinity binding mode, rather than adopting unstable or random orientations. Notably, the majority of the docking poses for these top candidates fall well below the reference ligand threshold (dashed line), confirming their robustness as potential inhibitors.

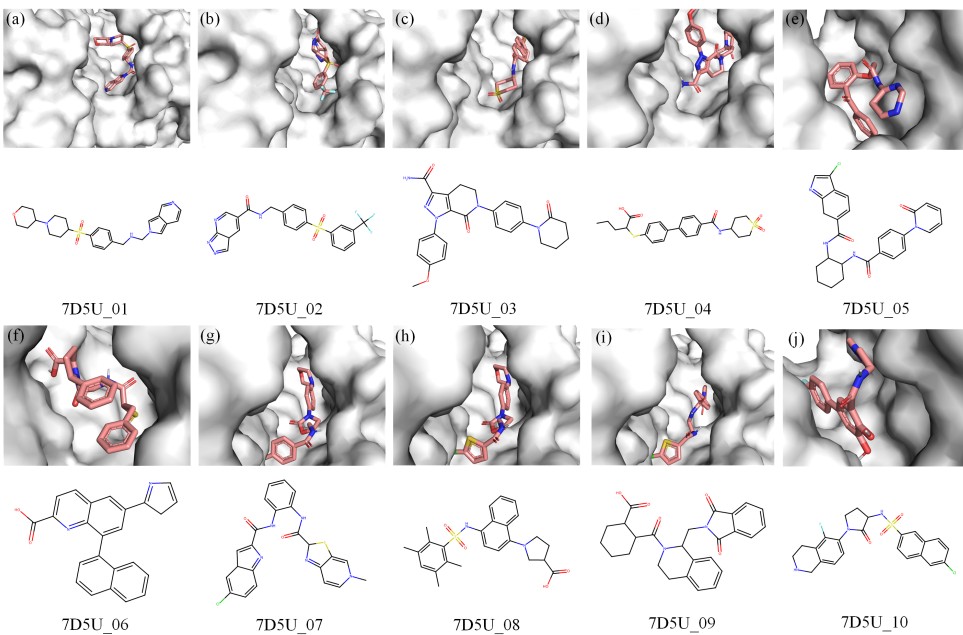

*Figure 7.* 3D and 2D visualizations of the ten ligand molecules docked to the 7D5U protein target.

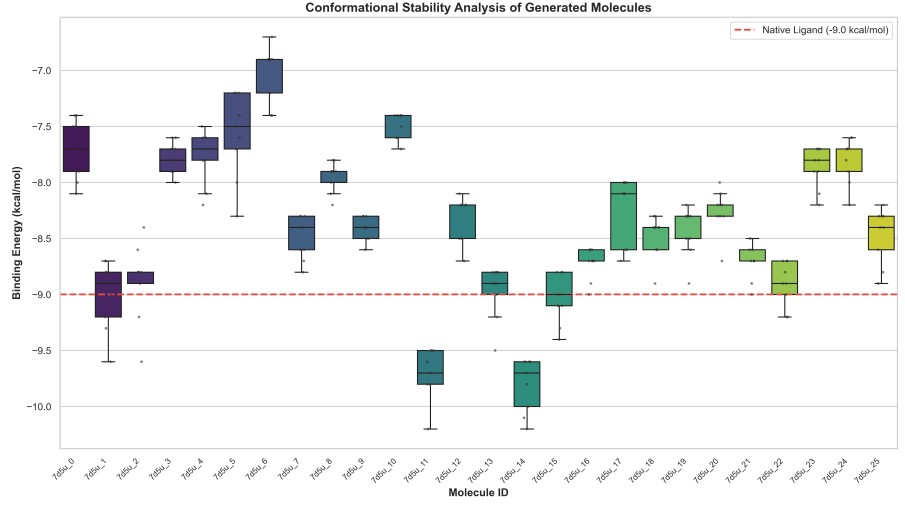

*Figure 8.* Docking energy landscape and conformational stability analysis of the generated molecules.

