# OpenReview forum: "TSMGen: Target-Specific Molecule Generation via Higher-Order Structural Dependencies and Context-Aware Bidirectional Fusion"
_ICML.cc/2026/Conference — ICML 2026 regular_

### Official Review · Reviewer_ySee · 2026-02-19

**Soundness:** 3
**Presentation:** 1
**Significance:** 3
**Originality:** 3
**Overall Recommendation:** 4
**Confidence:** 4

**Summary:**

Target-specific molecule generation is a core task in early-stage drug discovery, aimed at designing molecules that bind tightly to disease-related protein targets. Most existing generative methods for this task only model pairwise interactions between amino acids in the target pocket, failing to capture more complex higher-order spatial dependencies between groups of amino acids or atoms that mediate molecular recognition. To address this gap, this work proposes TSMGen, a target-specific molecule generation framework designed to model both local and global pocket structure by capturing higher-order spatial relations at both the amino acid residue and atomic levels. TSMGen also includes a context-aware bidirectional fusion module with gated cross-attention that jointly models features from both the protein pocket and the growing molecule during generation, rather than using the pocket as a static conditioning signal, to better align generated molecules with pocket binding constraints. The authors report that TSMGen outperforms existing baselines across multiple key metrics including binding affinity (Vina Score), high-affinity hit rate, QED (druglikeness), SA (synthetic accessibility), and structural diversity. A case study on the clinically relevant β-secretase target further demonstrates the model’s ability to generate high-affinity binders for real drug targets. The authors frame their core contributions as the integration of multi-scale higher-order pocket modeling, the bidirectional cross-attention fusion module, and strong empirical performance across public benchmarks.

**Compliance With Llm Reviewing Policy:**

Affirmed.

**Final Justification:**

In my initial assessment, I recognized the strong conceptual foundation and empirical potential of TSMGen. The approach addresses a critical gap in target-specific molecule generation by moving beyond pairwise interactions to capture higher-order spatial dependencies using hypergraphs, which is highly relevant for molecular recognition. However, my initial enthusiasm was significantly tempered by severe presentation issues that obscured the core methodology. Specifically, the manuscript was highly ambiguous about whether the framework was a de novo generator or a seed-based optimizer, it lacked a clear explanation of its generative paradigm, and it omitted comparisons to recent state-of-the-art baselines.

The authors' comprehensive rebuttal successfully resolved these fundamental ambiguities. They explicitly clarified that TSMGen is a conditional autoregressive de novo generation model that outputs discrete SMILES sequences, utilizing SMILES solely as a training signal rather than a required inference input. Furthermore, the inclusion of new benchmarking results against highly competitive baselines, namely MolCraft, TargetDiff, and UniMoMo, confirmed that TSMGen achieves genuine state-of-the-art performance across key metrics. The authors also provided a compelling distinction between their hypergraph approach and structure-prediction architectures like Evoformer, effectively contextualizing their design choices.

Based on these crucial clarifications and the newly provided empirical evidence, my final recommendation is a Weak Accept. The underlying method is technically sound, highly original, and demonstrates excellent performance, including a valuable case study on the BACE1 target. However, this positive recommendation is strongly contingent upon the authors fulfilling their commitment to fundamentally rewrite the methodology section and redesign the architectural figures. The core generative paradigm and exact input-output flow must be made explicitly clear early in the final manuscript to ensure the broader community can easily understand and build upon this promising work.

**Key Questions For Authors:**

1. The manuscript does not clearly specify the exact input–output interface of the generative pipeline. The input evidently includes the 3D structure of the target pocket. However, the model also appears to require a molecule SMILES string as input. Could the authors clarify the role of this molecular input? Is it used as a seed for autoregressive growth, as conditional context, or for training only?Regarding the output, what molecular representation does TSMGen generate? Is the model producing discrete SMILES sequences, 2D molecular graphs, or full 3D conformations?

2. Section 3.6 mentions the use of cross-entropy loss, but the underlying generative paradigm of TSMGen is not clearly stated. Could the authors clarify whether the model follows an autoregressive sequence generation framework, a discrete diffusion process, a variational formulation, a GAN-based approach, or another paradigm? Given that cross-entropy is typically associated with token-level likelihood maximization in autoregressive models, an explicit statement of the generative formulation and training objective would help situate TSMGen within the broader landscape of molecular generative modeling.

3. The authors emphasize modeling higher-order spatial relations among amino acids beyond pairwise interactions. In this context, it would be helpful to compare the proposed method with existed ones. For example, in protein structure modeling, architectures such as the Evoformer/Pairformer blocks in AlphaFold incorporate triangular updates to capture 3-residue (and higher-order) geometric constraints. Could the authors discuss the advantages of hypergraphs compared with existing high-order relation modeling methods like Pairformer?

**Limitations:**

Yes

**Strengths And Weaknesses:**

Strength

1. The core motivation addresses a well-documented gap in target-specific molecule generation: most existing methods only model pairwise amino acid/atomic interactions in protein pockets, ignoring higher-order spatial dependencies (e.g., coordinated 3-residue binding motifs) that are critical for molecular recognition.

2. The method have great performance, including both SOTA metric on benchmarks and the BACE1 case study in a clinically relevant use case, making the work accessible to both method developers and drug discovery.

Weakness

1. Critical ambiguity about the core conditioning setup: The paper repeatedly refers to "target-specific molecule generation" but never clarifies if TSMGen performs de novo generation using only the protein pocket as input, or requires an existing example/seed ligand as an additional input. This ambiguity invalidates assessment of the benchmarking: if TSMGen is a seed-based optimization method, comparing it to de novo generation baselines is unfair, as seed-based methods start with prior binding information and have a far lower performance bar. If seed ligands are required, the use of the CrossDock dataset is also unexplained, with no details on how seed ligands were sourced for training and testing, and no comparison to relevant molecular optimization baselines such as Delete.

2. Benchmarking is incomplete: Recent state-of-the-art target-specific de novo generation baselines including MolCraft, TargetDiff, and UniMoMo are omitted from experiments, making it impossible to confirm that TSMGen achieves true state-of-the-art performance, as the only baselines tested appear to be older or less competitive methods.

---

> ### Author Rebuttal · Authors · 2026-03-30
>
> We would like to thank the reviewers for their constructive feedback and provide the following responses to their comments.
>
> **1.Notes on input and output (Key Question 1 & Weakness 1):**
>
> We sincerely apologize for any ambiguity in our previous statement. We would like to clarify that TSMGen is a de novo molecular generation model conditioned on the protein pocket, rather than a seed-based optimization method. TSMGen requires SMILES strings as a supervisory signal only during the training phase. During the inference phase, the model generates molecules from scratch, requiring only the protein pocket as input and no seed ligand, thereby making the comparison fair with de novo design baseline models. TSMGen outputs discrete SMILES sequences, which can also be further converted into molecular diagrams and 3D conformations for evaluation.
>
> **2.Regarding the generative framework and training objectives (Key Question 2):**
>
> Thank you for allowing us to clarify this point. TSMGen follows a conditional autoregressive sequence generation framework, rather than a discrete diffusion, variational, or GAN-based paradigm. Conditioned on the protein pocket representation, the model generates SMILES tokens sequentially, where each token is predicted from the previously generated tokens under an autoregressive factorization. The training objective is a token-level cross-entropy loss, which is equivalent to maximum likelihood estimation. Therefore, TSMGen is fundamentally a conditional autoregressive molecular generation model.
>
> **3.Regarding the lack of recent baseline models (Weakness 2):**
>
> We thank the reviewer for the valuable suggestion. In response, we have added MolCraft, TargetDiff, and UniMoMo as additional baselines. The new results show that TSMGen consistently outperforms these baselines across the main metrics, further supporting its state-of-the-art performance, as shown in the table below.
>
> | Model | Vina Score (↓) | High Affinity (↑) | QED (↑) | SA (↑) | LogP | Lipinski (↑) | Diversity (↑) |
> | :--- | :---: | :---: | :---: | :---: | :---: | :---: | :---: |
> | MolCraft | -6.590 | - | 0.500 | 0.690 | - | - | 0.720 |
> | TargetDiff | -5.470 | 0.581 | 0.480 | 0.580 | - | - | 0.720 |
> | UniMoMo | -5.720 | - | 0.550 | 0.700 | 1.550 | 4.680 | - |
> | TSMGen | **-7.642** | **0.631** | **0.659** | **0.762** | **2.883** | **4.966** | **0.753** |
>
> **4.Comparison of higher-order modeling with Evoformer/Pairformer (Key Question 3):**
>
> We sincerely thank the reviewers for highlighting the connection between our hypergraph-based method and higher-order relations modeling approaches such as the Evoformer/Pairformer blocks in AlphaFold. The primary advantage of our hypergraph over Evoformer/Pairformer stems from the fundamental differences in their task objectives. Evoformer/Pairformer focuses on structure prediction, and such architectures are designed to map 1D sequences to 3D coordinates. They rely on dense triangular updates to enforce strict geometric constraints, which are essential for protein folding. In contrast, TSMGen focuses on conditional molecular generation; in our receptor-based generation task, the 3D structure of the pocket is already known a priori. Therefore, compared with triangular updates for global structure recovery, hypergraphs more directly capture multi-residue cooperative constraints within the pocket.

---

> > ### Author Rebuttal · Reviewer_ySee · 2026-04-01
> >
> > Thank you to the authors for the detailed rebuttal and for directly addressing my concerns. Your responses have successfully clarified the core mechanisms of TSMGen and resolved the fundamental ambiguities from the initial submission.
> >
> > Based on the clarifications and the new experimental results provided, I am raising my score to 4.
> >
> > Crucial Feedback for the Revision:
> >
> > While I am satisfied with the technical explanations provided in the rebuttal, the fact that the fundamental generative paradigm and input/output interfaces were this ambiguous indicates a major presentation issue in the current draft.I highly suggest a substantial rewrite of the methodology section to ensure these core details (SMILES-based autoregression, de novo pocket conditioning) are explicitly stated early in the text. Furthermore, please heavily modify your main architectural figures. They were very confusing upon initial reading; they must clearly and visually communicate the correct input-to-output flow so future readers do not experience the same misunderstandings.

---

> > > ### Author Response · Authors · 2026-04-01
> > >
> > > We sincerely thank you for reviewing our rebuttal and raising your score. We are deeply encouraged that our clarifications successfully resolved your concerns regarding the core mechanisms of TSMGen. We fully agree with your constructive feedback regarding the presentation of the manuscript, and we recognize that the current draft fell short in effectively communicating the input and output interfaces. In the final revision, we commit to substantially rewriting the methodology section so that the fundamental generative mechanism, specifically the SMILES-based autoregression and de novo pocket conditioning, is explicitly defined and emphasized early in the text. Furthermore, we will heavily modify our main architectural figures to visually and unambiguously communicate the correct flow from input to output, ensuring that future readers can immediately grasp the exact processes without experiencing the same confusion. Your insights have been invaluable in helping us improve the clarity and overall quality of our paper, and we sincerely appreciate your constructive guidance.

---

### Official Review · Reviewer_E5mm · 2026-03-11

**Soundness:** 3
**Presentation:** 3
**Significance:** 3
**Originality:** 3
**Overall Recommendation:** 5
**Confidence:** 4

**Summary:**

This paper proposes TSMGen, a molecule generation framework for targeted drug design by constructing residue-level hypergraphs and atomic-level graphs. This method comprehensively captures both local and global higher-order spatial structural dependencies within protein pockets. It also designs a context-aware bidirectional fusion module (CABF) that integrates protein pocket and molecular features via a dual-layer gated cross-attention mechanism. Experiments demonstrates that TSMGen achieves state-of-the-art performance on critical benchmarks like Vina Score, High Affinity, and QED, consistently outperforming baseline models. A case study on the Alzheimer's-related target β-secretase further shows the model's potential for real-world drug discovery.

**Compliance With Llm Reviewing Policy:**

Affirmed.

**Final Justification:**

I am raising my score from 4 to 5 after reading the rebuttal and considering the other reviews. The authors clarified a key ambiguity that the method is a de novo pocket-conditioned generator at inference time, not a seed-based optimization method, which addresses an fairness concern. They also added stronger recent baselines and provided useful clarifications on the evaluation protocol, synthesizability, validity, and computational cost. Overall, the rebuttal strengthened the paper.

**Key Questions For Authors:**

1.	While the Hypergraph Neural Network (HGNN) captures complex multi-residue interactions, hypergraph message passing is typically more computationally and memory intensive than standard pairwise graph convolutions. Could the authors provide a brief comparison of the training time and memory overhead between the HGNN and the standard atomic-level graph to clarify if the performance gains justify the additional computational cost?
2.	The framework captures features at different scales: atomic-level graphs for geometric details and residue-level hypergraphs for global interactions. Since residues are fundamentally composed of the exact same atoms, there is an inherent risk of overlapping structural signals. Could the authors clarify how the architecture ensures these two representations learn strictly complementary information rather than redundant features before reaching the CABF module?
3.	TSMGen demonstrates 3D pocket modeling capabilities but currently generates 1D SMILES strings. Given the recent success of 3D diffusion models in structure-based drug design, have the authors considered using this novel hypergraph representation as a conditional encoder for an end-to-end 3D diffusion framework? Discussing the technical feasibility of natively generating 3D atomic coordinates could provide valuable insights into future application.

**Limitations:**

The proposed framework demonstrates promising results in AI-driven drug design. However, the discussion of limitations could be further strengthened. The authors should emphasize the impact of their methodology on future studies and provide recommendations for further research. For instance, did the authors consider integrating domain knowledge resources such as the UMLS Metathesaurus to link each drug candidate to specific semantic types or biological pathways? The generalizability of the model to novel protein targets beyond the training distribution also remains unclear. Addressing these limitations in future work would enhance the real-world applicability of the approach.

**Strengths And Weaknesses:**

This paper proposes TSMGen, a structure-based molecular generation framework that integrates multi-scale representations of protein pockets, including atomic-level graphs and residue-level hypergraphs. By modeling higher-order residue relations through hypergraphs and introducing a context-aware bidirectional fusion module, the method aims to improve the interaction modeling between small molecules and protein pockets.
Strengths:
1.	Soundness: This work proposes a novel framework to advance drug design via integrating atomic-level graph representations with residue-level hypergraphs. A multi-scale pocket modeling approach captures both fine-grained atomic interactions and higher-order structural relations between residues. Rigorous ablation studies on key components validate their individual contributions to overall performance. Case studies support claims of improved drug-likeness. It provides a valuable perspective for de novo drug design.
2.	Presentation: The paper is generally well-written, with clear descriptions of the multi-scale pocket representation and the bidirectional fusion. The well-structured figures that effectively illustrate the model framework, making it easier to understand.
3.	Significance: Structure-based molecular generation remains an important and challenging problem in computational drug discovery. By incorporating multi-scale structural representations of protein pockets and modeling higher-order residue relationships, the proposed framework attempts to improve the modeling of ligand-pocket interactions. This direction is relevant to current research in AI-driven drug design and may provide useful insights for future studies on structure-aware molecular generation.
4.	Originality: The work combines atomic-level pocket graphs with residue-level hypergraphs and introduces a gated cross-attention based bidirectional fusion mechanism to integrate molecular and pocket representations. While the individual components are related to existing techniques, their integration into a multi-scale pocket modeling framework for molecular generation provides a reasonably novel perspective.
Weaknesses:
1.	The hypergraph construction relies on a fixed distance threshold (e.g., 5 Å), and the paper does not provide sufficient analysis of the sensitivity of the model to this parameter. Moreover, hypergraph neural networks may introduce additional computational overhead compared with standard graph representations, but the paper provides limited discussion on the training cost and scalability of the proposed framework.
2.	Hypergraph neural networks generally introduce additional computational overhead compared with standard graph representations. The paper would benefit from a more detailed discussion of training cost and scalability.
3.	The paper employs 1D SMILES sequences to represent molecules in feature representation. However, protein pockets are highly complex 3D spatial structures. Directly fusing 1D sequences with 3D pocket features may result in the loss of conformational constraints that govern molecules in their actual physical space.

---

> ### Author Rebuttal · Authors · 2026-03-30
>
> We would like to thank all the reviewers for the insightful comments and constructive suggestions.
>
> **1.Discussion on training cost, scalability, and memory overhead (Key question1 & Weakness 2):**
>
> We acknowledge the reviewer's concern that Hypergraph Neural Network (HGNN) generally introduce additional computational overhead and are more memory intensive compared with standard graph representations, as information propagation occurs through node-hyperedge-node pathways rather than simple pairwise interactions. However, in our framework, this additional computational cost is kept within strictly manageable limits for two main reasons. First, the number of residues within a protein pocket is naturally limited (typically ranging from several dozen to about a hundred residues), ensuring the scale of the hypergraph remains relatively small. Second, the hypergraph network in our implementation consists of only two layers, which ensures high scalability.
>
> **2.Justification for the fixed distance threshold (Weakness 1):**
>
> We agree that the hypergraph construction relies on a fixed distance threshold (5 Å); however, we would like to clarify that this is not an arbitrary hyperparameter. Rather, it is a deliberate design choice strictly based on rigorous biophysical and chemical principles. The vast majority of key non-covalent interactions that drive binding (such as hydrogen bonds and van der Waals forces) strictly occur within this spatial threshold.
>
> **3.Overlap and complementarity of multi-scale features (Key question2):**
>
> We thank the reviewer for this insightful question. We acknowledge that since residues are fundamentally composed of the exact same atoms, there is an inherent risk of overlapping structural signals. However, our architecture is specifically designed to ensure these two representations learn strictly complementary information rather than redundant features before reaching the CABF module. Although both representations originate from the same protein pocket structure, their feature extraction pathways are explicitly decoupled to capture features at different scales. The atomic-level graph focuses on fine-grained chemical and geometric details, such as atomic types, bonding relationships, and precise spatial coordinates, which are important for modeling local ligand–atom interactions. In contrast, the residue-level hypergraph captures higher-order relationships among multiple residues, reflecting the broader structural organization and cooperative effects within the binding pocket. Therefore, the architecture ensures that the atomic graph provides local interaction details while the residue hypergraph models global interactions and structural context, delivering strictly complementary inputs to the CABF module.
>
> **4.Prospects for Native 3D Generation from 1D SMILES (Key question3 & Weakness3):**
>
> Thank you for your suggestion. 3D diffusion models have recently demonstrated strong performance in structure-based drug design. As we mentioned in our response to reviewer HF5p, we chose SMILES-based molecular representations primarily to preserve the conformational flexibility of the generated molecules. However, we believe that end-to-end 3D diffusion frameworks represent a promising direction for future research. We plan to explore this approach in our future work to further improve the performance of structure-based molecular design.

---

> > ### Author Rebuttal · Reviewer_E5mm · 2026-04-01
> >
> > I am raising my score from 4 to 5 after reading the rebuttal and considering the other reviews. The authors clarified a key ambiguity that the method is a de novo pocket-conditioned generator at inference time, not a seed-based optimization method, which addresses an fairness concern. They also added stronger recent baselines and provided useful clarifications on the evaluation protocol, synthesizability, validity, and computational cost. Overall, the rebuttal strengthened the paper.

---

> > > ### Author Response · Authors · 2026-04-01
> > >
> > > Thank you very much for your time and dedication in reviewing our manuscript and the subsequent rebuttal. We sincerely appreciate the constructive suggestions and insightful feedback you have provided throughout this process.

---

### Official Review · Reviewer_84fF · 2026-03-13

**Soundness:** 2
**Presentation:** 2
**Significance:** 2
**Originality:** 2
**Overall Recommendation:** 4
**Confidence:** 3

**Summary:**

The paper proposes TSMGen, a target-specific molecular generation framework for structure-based drug design. The method represents protein pockets using both residue-level hypergraphs and atomic-level graphs, aiming to capture structural information at different scales. It then introduces a context-aware bidirectional fusion module with gated cross-attention to integrate pocket and molecular features for conditional molecule generation. Experiments on the CrossDocked benchmark show improvements over several baselines on metrics such as Vina Score, QED, SA, and Diversity, and a case study on β-secretase further suggests promising binding performance.

**Compliance With Llm Reviewing Policy:**

Affirmed.

**Final Justification:**

Authors' detailed and comprehensive response has generally addressed my concerns.

**Key Questions For Authors:**

1 Are the generated molecules always chemically valid? If not, could the authors report validity as an additional evaluation metric?

2 The current framework uses SMILES-based molecular representations. What is the rationale for this design choice, instead of representing molecules as graphs and learning with a GNN-based encoder/decoder?

3 For the hypergraph construction of protein pockets, have the authors compared the current radius-based method with a KNN-based method? Would KNN lead to more stable or better-performing pocket representations?

**Strengths And Weaknesses:**

Strength:

1 The paper presents comprehensive ablation studies, which help demonstrate the contribution of the major components.

2 The problem formulation is clear, well-motivated, and practically relevant for structure-based drug design.

Weakness:

1 The testing and evaluation protocol is not sufficiently clear. The paper states that “100 pairs from held-out clusters formed the test set,” but it is not fully explained how the results in Table 2 are computed. For example, it is unclear how many molecules are generated per protein pocket, how the reported metrics are aggregated, and whether the evaluation is performed per target or over all generated samples. In addition, the test set appears relatively small compared with the training set, which raises questions about the robustness of the evaluation.

2 The paper does not report the computational cost of training or inference, which makes it difficult to assess the practical efficiency and scalability of the method.

---

> ### Author Rebuttal · Authors · 2026-03-30
>
> We would like to thank all the reviewers for the insightful comments and constructive suggestions.
>
> **1.Regarding validity measures (Key Question 1):**
>
> Thank you for the suggestion. We agree that validity is an important metric for molecular generation. Our method achieves over 99% validity, indicating that the generated molecules largely satisfy chemical validity constraints. We will include this result in the revised version for completeness. Since validity is already very high, we mainly focus on more discriminative metrics such as binding affinity and drug-like properties.
>
> **2.Why choose SMILES over graph-based GNNs? (Key Question 2)**
>
> Thank you for your suggestion. We chose a SMILES-based molecular representation primarily to maintain conformational flexibility in the generated molecules. Using a GNN-based encoder/decoder would force the model to lock onto a rigid spatial conformation during the generation phase, whereas the SMILES notation prioritizes atomic connectivity over spatial configuration. This allows the generated molecules to dynamically adjust to the most favorable 3D pose when interacting with the target pocket.
>
> **3.A comparison of radius and KNN hypergraph construction methods (Key Question 3):**
>
> Thank you for pointing this out. We employ a radius-based approach to construct hypergraphs, aiming to more accurately capture the physical microenvironment of protein pockets, as biochemical interactions are strictly limited by spatial distance thresholds (e.g., a radius of r = 5 Å). However, the distribution of protein 3D structures is non-uniform. If a node is located on the sparse edge of a pocket, KNN will force connections across long physical distances to distant nodes in order to artificially meet the requirement of k neighbors. In biochemistry, no actual physicochemical forces (such as hydrogen bonds or van der Waals forces) exist between residues that are too far apart. Such forced connections introduce a significant amount of noise into the model.
>
> **4.Response to evaluation criteria (Weakness 1):**
>
> We agree with the reviewer on the importance of a highly transparent testing and evaluation protocol. To clarify how the results in Table 2 are computed: we generated 100 molecules per protein pocket, and the reported metrics are aggregated over all generated samples, rather than being computed per target. Regarding the concern that the test set appears relatively small, we would like to clarify that this division between training and testing, as well as the number of samples generated, is strictly consistent with the protocols used by other baselines. These 100 pairs were selected based on the standard data split adopted in CrossDocked2020. By adhering to this established benchmark setting, we ensure a fair, robust, and direct comparison with existing methods.
>
> **5.Response regarding computational cost and inference efficiency (Weakness 2):**
>
> Thank you for pointing this out. We agree that computational cost is important for assessing the efficiency and scalability of the method. We have now provided an analysis of the time complexity. The time complexity of the encoder processing pocket diagrams is $O(V + E)$, where $V$ represents the number of nodes in the protein pocket graph and $E$ represents the number of edges in the protein pocket graph. The time complexity of a single forward pass of the decoder’s self-attention mechanism is $O(L^2 \cdot d)$, where $L$ denotes the length of the SMILES string and $d$ represents the dimension of the feature embedding. During the inference stage, the time complexity of autoregressive generation is $O(L^3 \cdot d)$. Based on our practical testing, TSMGen takes approximately 10 hours to train on a single NVIDIA RTX 3090 GPU, and it takes an average of 10 seconds to generate 100 molecules during inference, while using less than 12 GB of VRAM.

---

> > ### Author Rebuttal · Reviewer_84fF · 2026-04-05
> >
> > Thank you for your detailed discussion and follow-up. Your response has generally addressed my concerns and I am willing to increase the score from 3 to 4

---

> > > ### Author Response · Authors · 2026-04-06
> > >
> > > Thank you for your continued engagement and the updated score. We sincerely appreciate your feedback, which has helped strengthen our work.

---

### Official Review · Reviewer_HF5p · 2026-03-13

**Soundness:** 3
**Presentation:** 3
**Significance:** 3
**Originality:** 3
**Overall Recommendation:** 4
**Confidence:** 4

**Summary:**

The authors propose a target specific molecule generation framework that captures local and global structural information from the protein pocket. They capture both residue level and atom level features with a context aware bidirectional fusion module that uses gated cross-attention between the protein pocket and the molecule. The generated molecules are measured using many metrics like docking score, qed and SA.

**Compliance With Llm Reviewing Policy:**

Affirmed.

**Final Justification:**

The authors have answered my questions and I retain my original score.

**Key Questions For Authors:**

(1) The synthetic accesibility scores are usually in the range of 1-10 with small numbers representing molecules that are easier to synthesize. Could you explain how your scores are from 0-1? Is this normalized?
(2) Could you provide what percentage of the top molecules can be easily synthesized with readily available components using a retrosynthesis tool
(3) Can you show the distribution of SA and QED scores?

**Limitations:**

yes

**Strengths And Weaknesses:**

Strengths:

(1) They use multiple resolution representation for protein which can better capture the protein pocket.
(2) Using hypergraphs for capturing residue interactions is novel
(3) cross attention fusion is also novel in this context.

Weaknesses:
(1) There are no details on how the generated molecules can be synthesized as generative models are prone to suggesting molecules that are not possible to synthesize. A characterization of how many molecules fron the generated set can be synthesized in fewer than 3-4 steps using starting materials easily available would be good.
(2) Some simple algorithms that are likely to perform well like genetic algorithms are not part of the baseline (See Tripp et al 2023 https://arxiv.org/abs/2310.09267)

---

> ### Author Rebuttal · Authors · 2026-03-30
>
> We sincerely thank the reviewer for the constructive feedback on our paper and for recognizing the novelty of our hypergraph mechanism and context-aware bidirectional fusion module. Below is our point-by-point response to your specific comments.
>
>  **1.Clarification on the range and normalization of the SA score (Key Question 1):**
>
> Thank you for pointing this out. You are correct that the original SA scores typically range from 1 to 10, with 1 indicating that the compound is extremely easy to synthesize. In our study, to maintain consistency with the SA results reported by other compared methods, we normalized the original SA scores to the range of [0,1] using the following transformation formula:
> $sa_{\text{norm}} = \mathrm{round}\left(\frac{10 - sa}{9}, 2\right)$.
>
> **2.Details on synthetic accessibility and retrosynthetic analysis (Weakness 1 & Key Question2):**
>
> We thank the reviewer for this valuable suggestion. We agree that the practical synthesizability of deep generative models is a key challenge. Following this suggestion, we have added a new experiment to further assess the practical synthesizability of the generated molecules. We used the inverse synthesis planning tool AiZynthFinder to evaluate the 50 generated molecules for the 7D5U target in the case study. The results show that 30% of the generated molecules are predicted to be synthesizable in 3-4 steps, 60% in 5-6 steps, and 10% require 7 or more steps.
>
> **3.Supplement genetic algorithms baseline (Weakness 2):**
>
> We thank the reviewers for pointing out the insightful work of Tripp et al. (2023). We agree that simple algorithms such as genetic algorithms perform well in molecular generation tasks. However, this study strictly focuses on 3D receptor-conditioned molecular generation, where the primary challenge lies in modeling the complex spatial dependencies within protein pockets. Therefore, we selected our baselines primarily because they are capable of natively handling 3D pocket structures. In this domain, genetic algorithms lack native mechanisms for handling 3D spatial constraints, making it difficult to directly transfer them to such structured tasks. Therefore, we did not include them in the current benchmark, but we will consider them in future work.
>
> **4.Distribution of SA and QED (Key Question 3):**
>
> We evaluated the SA and QED distributions of the generated molecules. For SA, the average normalized SA score was 0.762, with over 92.8% of the candidate molecules scored above 0.6, over 64% scored above 0.7, and less than 1% scored below 0.5. For QED, the average score was 0.659, with the highest score reaching 0.934. Notably, over 65% of the candidate molecules scored above 0.5, and nearly 30% scored above 0.7.

---

### Decision · Program_Chairs · 2026-04-30

**Decision:**

Accept (regular)

**Comment:**

This paper proposes a pocket-conditioned small molecule generation framework, introducing a residue-level hypergraph with gated cross-attention for higher-order pocket interaction modeling, coupled with an autoregressive SMILES decoder. The hypergraph encoding and bidirectional fusion modules are technically sound and address meaningful gaps in structure-based de novo generation. The model also showed strong empirical performance across standard benchmarks.

**Rebuttal summary**
- Concerns raised on ligand synthesizability, validity, cost efficiency, and missing updated baselines. Authors adequately addressed with additional analysis on synthesizability and computational cost. With additional methodology clarifications, initial reviewer confusions are resolved.
- However, I found the response to SMILES vs. conformer representation choice is weak and generalizability to novel protein targets remain partially unresolved.

**Recommendation: Weak Accept** A novel hypergraph contribution with strong empirical support. If accepted, authors should include the generalizability discussion and revise the methodology section and Figure 1 for clarify.